# Rift Valley Fever Outbreak Investigation Associated with a Dairy Farm Abortion Storm, Mbarara District, Western Uganda, 2023

**DOI:** 10.3390/v17071015

**Published:** 2025-07-19

**Authors:** Luke Nyakarahuka, Shannon Whitmer, Sophia Mulei, Joanita Mutesi, Jimmy Baluku, Jackson Kyondo, Amy Whitesell, Carson Telford, Alex Tumusiime, Calvin Richie Torach, Dianah Namanya, Mariam Nambuya, Dominic Muhereza, Zainah Kabami, Annet Nankya, David Muwanguzi, Francis Mugabi, Nelson Wandera, Rose Muhindo, Joel M. Montgomery, Julius J. Lutwama, Stephen Karabyo Balinandi, John D. Klena, Trevor R. Shoemaker

**Affiliations:** 1Uganda Virus Research Institute, Entebbe P.O. Box 49, Uganda; smulei@uvri.go.ug (S.M.); jmutesi@uvri.go.ug (J.M.); jbaluku@uvri.go.ug (J.B.); jkyondo@uvri.go.ug (J.K.); atumusiime@uvri.go.ug (A.T.); ctorach@uvri.go.ug (C.R.T.); dnamanya@uvri.go.ug (D.N.); jlutwaama@uvri.go.ug (J.J.L.); sbalinandi@uvri.go.ug (S.K.B.); 2Department of Biosecurity, Ecosystems and Veterinary Public Health, Makerere University, Kampala P.O. Box 7062, Uganda; 3US Centers for Disease Control and Prevention, Atlanta, GA 30329-4027, USA; evk3@cdc.gov (S.W.); nmm9@cdc.gov (A.W.); pwv0@cdc.gov (C.T.); ztq9@cdc.gov (J.M.M.); irc4@cdc.gov (J.D.K.); tis8@cdc.gov (T.R.S.); 4Ministry of Health of Uganda, Kampala P.O. Box 7272, Uganda; mariamnambuya@gmail.com (M.N.); daudimuwanguzi@gmail.com (D.M.); 5Mbarara Regional Referral Hospital, Mbarara P.O. Box 40, Uganda; dominy2468@gmail.com (D.M.); fmugabi2004@yahoo.com (F.M.); dennisnelson000@gmail.com (N.W.); 6Public Health Fellowship Program, Uganda National Institute of Public Health, Kampala P.O. Box 7272, Uganda; zkabami@uniph.go.ug (Z.K.); anankya@uvri.go.ug (A.N.); 7Department of Internal Medicine, Faculty of Medicine, Mbarara University of Science and Technology, Mbarara P.O. Box 1410, Uganda; drmuhindo@gmail.com

**Keywords:** Rift Valley Fever, zoonotic transmission, animal abortion storms, human RVF outbreak, western Uganda, livestock–human interface, RVF virus, vaccination trials

## Abstract

In Africa, Rift Valley Fever poses a substantial risk to animal health, and human cases occur after contact with infected animals or their tissues. RVF has re-emerged in Uganda after nearly five decades, with multiple outbreaks recorded since 2016. We investigated a unique RVF outbreak associated with an animal abortion storm of 30 events and human cases on a dairy farm in Mbarara District, Western Uganda, in February 2023. Genomic analysis was performed, comparing animal and human RVF viruses (RVFV) circulating in the region. A cluster of thirteen human RVF cases and nine PCR-positive animals could directly be linked with the abortion storm. Overall, during the year 2023, we confirmed 61 human RVFV cases across Uganda, 88.5% of which were reported to have had direct contact with livestock, and a high case fatality rate of 31%. We recommend implementing extensive health education programs in affected communities and using sustainable mosquito control strategies to limit transmission in livestock, coupled with initiating animal vaccination trials in Uganda.

## 1. Introduction

Rift Valley Fever (RVF) has emerged as a significant One Health concern in Uganda, marked by the first laboratory-confirmed outbreak in 48 years in Kabale District in 2016 [1]. Subsequent studies demonstrated circulation of the RVF virus (RVFV) in both humans and animals, with a seroprevalence of 12% in humans and 13% in animals in the Kabale area [2]. Risk factors for RVFV infections include the consumption of RVF-infected animal products, contact with RVF-infected animals, and residence in low-altitude areas conducive to vector emergence [3].

In a recent publication by Nyakarahuka et al. (2023), 52 human cases of laboratory-confirmed RVF were reported from 2017 to 2020, with a case fatality rate (CFR) of 42% [4]. Most cases originated from central and western districts, part of the cattle corridor of Uganda, where direct contact with livestock emerged as a primary risk factor. In-depth sequencing analysis showed that two clades of RVFV are circulating in Uganda [4]. The Kenya-2 clade, previously detected in Kenya and Sudan, was the first clade identified in Uganda and continues to circulate [4]. A second, distinct clade, the “K/E” clade, was first identified in Uganda in 2018 [4].

The Uganda Virus Research Institute (UVRI) continues to conduct surveillance for viral hemorrhagic fevers (VHF), including RVF, as part of the Integrated Disease Surveillance and Response plan (IDSR) of WHO. In February 2023, human VHF-suspect samples were submitted to UVRI following spontaneous animal abortions on a dairy farm in Mbarara District, western Uganda. One sub-county (Rwanyamahembe) in this district was the focus of subsequent human case detection. In this study, we detail the investigation involving both human and animal cases, along with laboratory confirmation, including metagenomic Next-Generation Sequencing (mNGS) analyses.

## 2. Materials and Methods

### 2.1. Case Detection and Field Investigation

Cases were identified through UVRI’s VHF Surveillance Program [5]. In February 2023, human blood samples from suspect cases meeting the VHF case definition were received from several Mbarara district health facilities, primarily from Bwizibwera Health Centre IV. The case definition used for collection of a sample for submission to UVRI involved any person presenting with an undiagnosed fever of greater than or equal to 38.0 °C, with or without any bleeding, and lacking an alternative diagnosis. Cases of RVF were confirmed by real-time reverse transcription polymerase chain reaction (rRT-PCR) testing. Collaborating with district health and veterinary teams, UVRI, the Viral Special Pathogens Branch (VSPB) of the US Center for Disease Control and Prevention (CDC), and other partners conducted investigations in Mbarara District, focusing on the most affected subcounty of Rwanyamahembe.

Initial outbreak assessment in Rwanyamahembe sub-county revealed more cases within the community, particularly among individuals associated with providing animal care on one dairy farm. Additional samples were collected from herdsmen and animals on this dairy farm. The outbreak investigation was broadened to include surrounding villages and neighboring farms (Figure 1), collecting samples from symptomatic individuals and those who had contact with suspected animals. Whole blood (EDTA) samples were obtained from humans and animals; these samples were tested at UVRI using a previously published algorithm consisting of rRT-PCR and IgM/IgG ELISA testing [6]. The investigation also involved the collection of epidemiological data, including variables associated with RVF transmission, such as livestock contact and mosquito density in the area. Data were captured digitally using a standardized questionnaire in Epi Info™ software (version 7, U.S. Centers for Disease Control and Prevention, Atlanta, GA, USA) [7]. 

### 2.2. Laboratory Methods

RNA was extracted using a 5X Magmax™ 96 Viral Isolation kit (Applied Biosystems Inc., Vilnius, Lithuania), and rRT-PCR was performed using a VSPB laboratory-developed test as previously published [6]. Samples containing RVFV-reactive IgG and IgM antibodies were evaluated using two VSPB laboratory-developed ELISA assays [6].

### 2.3. High-Throughput Sequencing, Bioinformatics, and Phylogenetics

RNA was extracted from blood at the UVRI VHF Program Laboratory and CDC’s VSPB laboratory in Atlanta, GA, using Magmax™ 96 Viral Isolation kits (Applied Biosystems Inc.) and readied for sequencing using either an unbiased library preparation (Illumina) or an RVFV-specific amplicon library preparation (MinIon). For unbiased Illumina library preparation, RNA was DNase-treated, and libraries were prepared using the NEBNext Ultra II (New England Biolabs, Ipswich, MA, USA). directional RNA library kit and sequenced using either the Illumina iSeq™ 100 (V1, 300-cycle kit) or MiSeq™ (V2, 300-cycle kit) systems (Illumina, San Diego, CA, USA) [4,8]. RVFV samples were sequenced using RVFV-specific primers designed using Uganda-specific RVFV sequences from Nyakarahuka et al. (2023) [4]. Consensus genome sequences were constructed using the bioinformatics method appropriate to the library construction method—either the ARTIC bioinformatics protocol (for amplicon-based MinIon sequencing) using RVFV-specific config files or a read mapping to a reference genome sequence using in-house scripts (https://github.com/evk3 accessed on 18 November 2024) for TruSeq-based Illumina sequencing. For TruSeq-based Illumina sequencing, low-quality reads/bases were filtered using Prinseq-lite v0.20.3 (-min_qual_mean 25-trim_qual_right 20-min_len 50), and RVFV genome sequences were assembled by aligning trimmed reads to an RVFV Uganda sequence from 2017 (MG972978-80) using BWA-mem (29) and iteratively mapped to the intermediate scaffold genome sequence; new consensus genome sequences were called using samtools mpileup (-A-aa-d 6000000-B-Q 0) and ivar consensus (1.3.1) (-m 2-n N). A megablast of assembled contigs was also used to confirm the closest matching RVFV reference sequences located in GenBank. Phylogenetic trees, based on full-length S, M, and L RVFV segments, were constructed using RAxML, visualized with ggtree, and clades labeled [4]. The inferred evolutionary relatedness of all known full-length RVFV sequences from Uganda was analyzed using the virus L, M, and S segment sequences. L, M, and S segments were aligned using MAFFT (v 7.450, Kazutaka Katoh, Osaka, Osaka Prefecture, Japan), and trees were built with RAXML-ng (v 1.2.2, -model GTR+G--seed $RANDOM--threads 16--bs-metric fbp,tbe). The final tree figure was generated using the ggtree library (v 3.2.0, Bioconductor, Seattle, WA, USA) in R (v 4.1.3, R Foundation for Statistical Computing, Vienna, Austria) with JupyterLab (v 3.4.5, Project Jupyter, New York, NY, USA). Genomes were deposited into GenBank, accession numbers PV849477–PV849536.

### 2.4. Data Analysis and Management

Data were captured in EpiInfo using a standardized VHF data capture form designed for surveillance [9]. Animal data were collected on an animal data collection form designed in EpiInfo. After several rounds of data cleaning, data were transformed into MS Excel (Version 16.88) and later imported into RStudio (Version 2023.06.0) for analysis. Univariate and bivariate data analyses were conducted in RStudio [10]. Coordinates collected during the investigation were utilized to map the location of cases using QGIS (version X) [11].

## 3. Results

In 2023, an outbreak of RVFV was observed in humans in Mbarara District, Uganda. Epidemiological investigations identified 61 laboratory-confirmed human cases, with 80% of these cases geographically concentrated in the Ankole sub-region, predominantly in the greater Mbarara District (Figure 1). Temporally, 64% of the cases occurred between January and April 2023 (Figure 2), with Mbarara District as the epicenter. Occupational risk analysis revealed that 75% of the cases were linked to the livestock business, involving farmers, herdsmen, and butchers (Table 1). Direct contact with livestock was reported in 88.5% of the cases (54/61), strongly suggesting a zoonotic transmission pathway. Demographically, 91.8% of the cases were male, with a median age of 34 years (ranging from 27 to 48 years). Adults constituted 98% of the cases, with only one child affected. The most prevalent clinical symptom was fever (69%), followed by joint pain (56%), headache (56%), muscle pain (49%), and anorexia (49%). Other frequently reported symptoms included abdominal pain (38%), unexplained bleeding (41%), diarrhea (18%), cough (11%), and skin rash (3.3%). The case fatality rate (CFR) was notably high at 31% (19/61 cases), underscoring the severity of the outbreak.

A Pearson’s chi-squared analysis revealed significant gender differences, with 90.2% of RVF-positive cases being male compared to 65.1% of RVF-negative cases (*p* < 0.001) (Table 2). Age group distributions were not significantly different, but the highest proportion of laboratory-confirmed cases was within the 40–54 years age group (39.3%). Clinical symptoms were markedly more prevalent in positive cases, with fever (68.9%), unexplained bleeding (41.0%), vomiting/nausea (45.9%), intense fatigue/general weakness (60.7%), and muscle pain (49.2%) being significantly higher compared to negative cases (*p* < 0.001).

### 3.1. Investigation of Abortion Storms at Dairy Farm and Potential Link to Human Cases

One dairy farm experienced a notable history of cattle abortions (Figure 2), recording 30 cases from December 2022 to March 2023. In 2023, 21.3% (13/61) of human cases were detected on this dairy farm, particularly in January and February, coinciding with the reported animal abortions. To understand the potential link, a thorough animal sampling was conducted from 47 animals (Table 3); 38 of these animals were collected from the dairy farm. The sampled animals were primarily adult cattle (89.4%), predominantly female (85.1%), accompanied by a smaller percentage of sheep (10.6%). Among the sampled animals, six (6) were reported to have had spontaneous abortions at the time of sampling. Results from rRT-PCR testing revealed that 21.3% (10 out of 47) of the animals tested positive for RVFV, with nine (9) positives identified at this particular dairy farm. Serological screening revealed that 57.4% (27/47) of the animals were RVFV IgG-positive, including 22 positives from the same dairy farm. No statistically significant differences for RVFV seropositivity were observed for breed type or animal species (Table 3). It was noted that there was a statistically significant difference between different age groups, with adults being more seropositive (*p* = 0.032), and also female animals being more seropositive (*p* = 0.027).

### 3.2. Results of the Next-Generation Sequencing and Phylogenetic Analysis of the Mbarara RVF Outbreak Cluster

These sequences were classified as members of the Kenya-2 clade, showing close evolutionary relationships to other East African RVFV sequences, including those from Kenya in 2007 and Sudan in 2010. The RVFV sequences from cattle and humans in the Mbarara region (highlighted in blue in Figure 3) exhibited tight clustering, strongly suggesting interspecies transmission during the same timeframe (Figure 3, Appendix A). Interestingly, two RVFV sequences collected from Rubanda District (southwestern Uganda, Rubanda) in 2023 were genetically distinct from the Mbarara outbreak cluster. These Rubanda sequences were closely related to RVFV genomes collected from Uganda in 2018, suggesting a viral spillover event from a different source than the Mbarara outbreak (Figure 3, Appendix A).

## 4. Discussion

The 2023 Rift Valley Fever (RVF) outbreak in Mbarara, Uganda, represents a significant public health and veterinary concern, reinforcing the increasing frequency of zoonotic spillover events in East Africa. This outbreak, which resulted in 61 laboratory-confirmed human cases, was concentrated in the cattle corridor of western Uganda, a critical livestock trade hub. The high case fatality rate of 31% is markedly higher than the global average of 1–10%, suggesting either severe disease manifestations or underreporting of mild cases [3]. The strong occupational association with livestock handling, with 88.5% of cases reporting direct contact with livestock, aligns with existing literature highlighting farming, herding, and slaughterhouse activities as key risk factors for zoonotic transmission [14].

Temporal clustering of cases between January and April 2023 aligns with established RVF epidemiology, where outbreaks peak during wet seasons due to increased mosquito activity and virus amplification [15]. Similar outbreaks in Kenya (2007), South Africa (2018), and Mauritania (2010) followed periods of heavy rainfall, reinforcing climate variability as a critical determinant of RVF transmission [16,17]. The phylogenetic analysis identified the Kenya-2 clade, which has circulated in Kenya, Sudan, and Tanzania, indicating regional viral persistence and cross-border movement [18]. However, the detection of distinct RVFV strains in Rubanda District suggests potential viral evolution and adaptation, necessitating enhanced genomic surveillance [19].

A key epidemiological finding was the link between livestock abortion storms and human cases. A dairy farm that reported 30 spontaneous cattle abortions between December 2022 and March 2023 accounted for 21.3% (13/61) of human infections. RVFV RNA detection in 9 out of 47 livestock samples supports direct zoonotic transmission. This mirrors previous outbreaks in Mauritania, South Africa, and Kenya, where large-scale livestock abortions preceded human cases, underscoring the potential role of reproductive anomalies in livestock as an early warning system [20,21]. Additionally, geostatistical models have demonstrated that livestock seroprevalence data can predict human case emergence, emphasizing the necessity for integrated livestock surveillance [22].

The observed gender disparity, with 91.8% of cases occurring in males, aligns with studies from Kenya, Sudan, and Mayotte, where male livestock farmers and butchers faced higher occupational exposure risks [23]. Age distribution analysis, with a median of 34 years and 98% of cases being adults, suggests that occupational exposure, rather than inherent age-related susceptibility, is the primary risk factor [14]. This pattern is consistent with findings from Sudan and Kenya, where younger individuals had lower infection rates due to reduced engagement in high-risk livestock handling activities [23].

Clinically, the most prevalent symptoms—fever (69%), joint pain (56%), unexplained bleeding (41%), and muscle pain (49%)—are consistent with previous RVF outbreaks in Mayotte (2018–2019) and Uganda (2017–2020) [4,20,24]. The significantly higher prevalence of unexplained bleeding among RVF-positive cases (*p* < 0.001) highlights its role as a clinical hallmark of severe RVF. The exceptionally high CFR of 31% is concerning, as previous Sudan (2010) and Kenya (2007) outbreaks reported CFRs below 15%, suggesting enhanced viral virulence or preferential detection of severe cases due to diagnostic limitations [25].

From a public health perspective, the outbreak underscores the urgency of enhancing RVF surveillance in Uganda. The current passive surveillance system, which relies on symptomatic case reporting, is insufficient for detecting early transmission events [26]. Implementing sentinel surveillance, incorporating routine livestock serological testing in high-risk areas, could facilitate early interventions before human cases emerge [27]. Climate models predict increasing RVFV outbreaks in East Africa due to shifting rainfall patterns, necessitating the development of forecasting models for Uganda [22].

Livestock vaccination is a critical strategy for outbreak prevention. Although live-attenuated RVFV vaccines exist, their use in Uganda has been limited due to logistical and regulatory challenges [25]. Kenya has demonstrated that vaccination campaigns significantly reduce human infections, underscoring the need for Uganda to integrate RVF vaccination into national veterinary health programs. Vector control measures, including insecticide-treated livestock shelters and environmental modifications, are essential in reducing mosquito populations and limiting viral transmission.

Public health risk communication and community engagement are equally vital. Many farmers and butchers in Uganda remain unaware of RVF transmission risks, leading to unsafe practices such as handling aborted fetal materials without protective measures [28,29]. Community-based education programs in Kenya and Sudan have demonstrated the effectiveness of targeted awareness campaigns in reducing occupational RVF exposure [30].

The 2023 RVF outbreak in Mbarara, Uganda, highlights the persistent threat of RVFV transmission in East Africa and underscores the urgent need for a multisectoral One Health approach. Given the strong correlation between livestock abortion storms and human cases, integrating veterinary and human health surveillance systems is imperative. Uganda must prioritize RVF vaccination trials, implement livestock abortion surveillance, expand genomic monitoring, and strengthen community education initiatives to mitigate future outbreaks and prevent zoonotic spillover events.

## Figures and Tables

**Figure 1 viruses-17-01015-f001:**
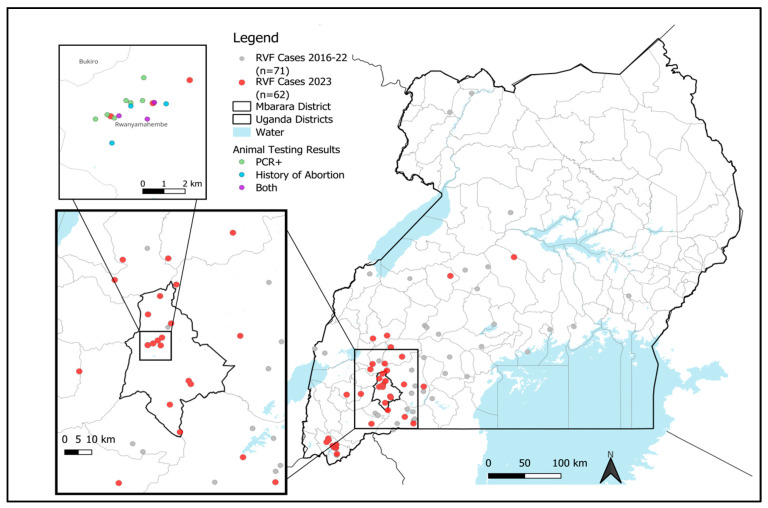
A map of Uganda showing the location of the 2023 Mbarara RVF outbreak and laboratory-confirmed relative to historic RVFV cases.

**Figure 2 viruses-17-01015-f002:**
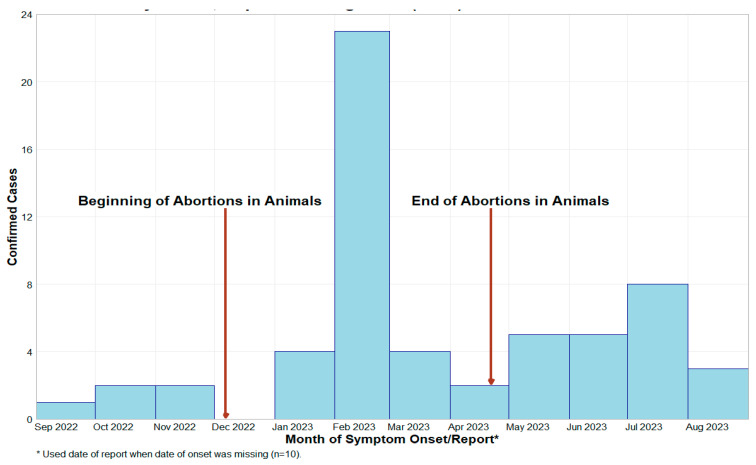
Epidemic curve of laboratory-confirmed human RVF cases in Uganda from September 2022 to August 2023.

**Figure 3 viruses-17-01015-f003:**
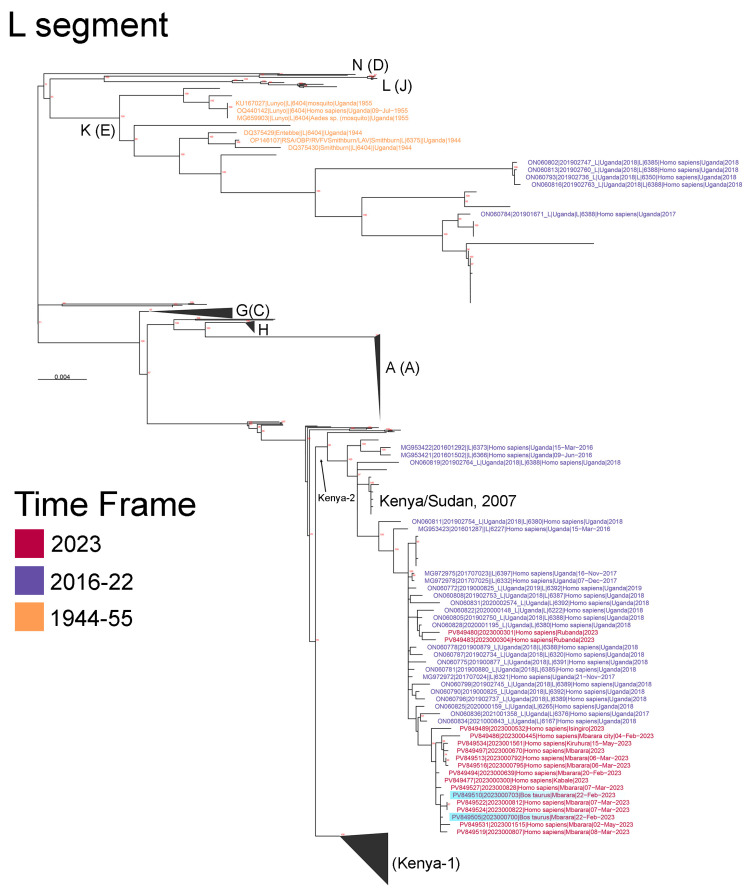
Phylogenetic analysis of the L segment of the Rift Valley Fever virus sequences in Uganda across different time frames (1944–2023). Sequences collected from Uganda are labeled, with sequence names color-coded by their collection timeframe. Blue shading highlights RVFV sequences collected from cattle. Notably, sequences from humans and cattle during the Mbarara outbreak and surrounding regions were closely related, suggesting zoonotic transmission. Bootstrap support values greater than 70% are labeled in red at internal nodes. Major clades are labeled according to Grobbalaar et al. (2011) [12] and Samy et al. (2017) [13], and the scale bar is in units of substitutions/site.

**Table 1 viruses-17-01015-t001:** Demographic characteristics and clinical symptoms of RVF cases detected in Uganda, January–December 2023.

Demographic Characteristics	N = 61 (%)	Clinical Symptoms	N = 61 (%)
RVF Cases	61 (100%)	Fever	42 (69%)
Female	5 (8.2%)	Unexplained bleeding	25 (41%)
Male	56 (91.8%)	Fatigue/general weakness	37 (61%)
Status		Anorexia	30 (49%)
Alive	42 (68.9%)	Abdominal pain	23 (38%)
Dead	19 (31.1%)	Headache	34 (56%)
Region		Vomiting/nausea	28 (46%)
Central	2 (3.3%)	Diarrhea	11 (18%)
Northern	1 (1.6%)	Muscle pain	30 (49%)
Western	58 (95.1%)	Joint pain	34 (56%)
Surveillance Type		Chest pain	18 (30%)
Active	21 (34.4%)	Conjunctivitis	12 (20%)
Passive	40 (65.6%)	Sore throat	4 (6.6%)
Contact with Livestock	53 (86.9%)	Difficulty breathing	10 (16%)
Travel history	10 (16.4%)	Difficulty swallowing	5 (8.2%)
Cattle Corridor	50 (81.7%)	Hiccups	4 (6.6%)
Age, years		Cough	7 (11%)
0–19	4 (6.6%)	Skin rash	2 (3.3%)
20–49	46 (75.4%)		
50>	11 (18%)		

**Table 2 viruses-17-01015-t002:** Comparative analysis of RVF-positive and RVF-negative cases: demographic characteristics, clinical symptoms, and associated risk factors.

Characteristic	Positive (N = 61)	Negative (N = 63)	*p*-Value ^1^	RR (95% CI)
Gender			<0.001	
Female	5 (8.2%)	22 (34.9%)		Reference
Male	56 (91.8%)	41 (65.1%)		2.67 (1.29–5.55)
Age Group			0.2	
<25	11 (18.0%)	12 (19.0%)		Reference
25–39	15 (24.6%)	18 (28.6%)		0.95 (0.54–1.68)
40–54	24 (39.3%)	14 (22.2%)		1.32 (0.81–2.16)
55+	11 (18.0%)	19 (30.2%)		0.77 (0.41–1.45)
Farmer	23 (37.7%)	23 (36.5%)	0.9	1.03 (0.71–1.48)
Butcher	10 (16.4%)	17 (27.0%)	0.2	0.70 (0.42–1.19)
Healthcare worker	1 (1.6%)	1 (1.6%)	0.9	1.02 (0.25–4.11)
Fever	42 (68.9%)	10 (15.9%)	<0.001	3.06 (2.04–4.60)
Unexplained bleeding from any site	25 (41.0%)	3 (4.8%)	<0.001	8.5 (2.7–26.8)
Vomiting/Nausea	28 (45.9%)	2 (3.2%)	<0.001	14.2 (3.5–57.7)
Bleeding of the gums	2 (3.3%)	0 (0%)	0.2	∞ (0.92–∞)
Diarrhea	11 (18.0%)	1 (1.6%)	0.002	11.4 (1.5–86.0)
Bleeding from injection site	2 (3.3%)	0 (0%)	0.2	∞ (0.92–∞)
Intense fatigue/General weakness	37 (60.7%)	7 (11.1%)	<0.001	2.80 (1.96–4.01)
Abdominal pain	23 (37.7%)	13 (20.6%)	0.036	1.48 (1.05–2.09)
Chest pain	18 (29.5%)	2 (3.2%)	<0.001	9.2 (2.2–37.8)
Muscle pain	30 (49.2%)	6 (9.5%)	<0.001	2.37 (1.72–3.25)
Joint pain	34 (55.7%)	10 (15.9%)	<0.001	2.29 (1.62–3.24)
Headache	34 (55.7%)	17 (27.0%)	0.001	1.80 (1.26–2.58)
Cough	7 (11.5%)	5 (7.9%)	0.5	1.21 (0.72–2.03)
Contact with Livestock	53 (86.9%)	53 (84.1%)	0.7	1.13 (0.65–1.95)

^1^ Pearson’s chi-squared test; Fisher’s exact test; RR = risk ratio.

**Table 3 viruses-17-01015-t003:** Characteristics of animals tested for RVFV by rRT-PCR and IgG ELISA.

Characteristic	Positive on IgG ELISA, N = 27	Negative on IgG ELISA, N = 20	*p*-Value
Breed			
Cross	13 (48%)	11 (55%)	0.5
Exotic	12 (44%)	6 (30%)	
Local	2 (7%)	3 (15%)	
Species			
Cattle	25 (93%)	17 (85%)	0.6
Sheep	2 (7%)	3 (15%)	
Sex			
Female	27 (100%)	16 (80%)	*0.032*
Male	0 (0%)	4 (20%)	
Age			
Adult	26 (96%)	14 (70%)	*0.027*
Young	1 (4%)	6 (30%)	
Abortions	5 (19%)	1 (5%)	0.2
PCR Results			
Negative	21 (78%)	15 (75%)	0.9
Positive	6 (22%)	5 (25%)	
Associated with human cases	17 (63%)	10 (50%)	0.4

## Data Availability

The original contributions presented in this study are included in the article/Appendix A. Further inquiries can be directed to the corresponding author.

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
