# Peer review of "Rift Valley Fever Outbreak Investigation Associated with a Dairy Farm Abortion Storm, Mbarara District, Western Uganda, 2023"

_viruses, 2025, doi:10.3390/v17071015_

Round 1
Reviewer 1 Report
Comments and Suggestions for Authors
his paper presents the findings of an epidemiological study conducted during a 2023 outbreak of RVF in Mbarara District, Uganda.
Interestingly, this outbreak had a higher CF ratio than is typically associated with the disease, which suggests that many cases may go unnoticed or be misdiagnosed. As with other outbreaks in different African regions, the zoonotic origin of the disease remains consistent, further reinforcing the need for a holistic health approach to prevent outbreaks and control the spread of the disease.
While there is a general consensus on the need for preventive detection and control systems, they are difficult to implement in affected countries, so the disease is likely to persist as a significant challenge.
The methods and results are clearly written, and the article is easy to read.
Specific comments:
The article provides information on the 2023 outbreak. It might be interesting to know the authors' view on the long period with no reporting of disease in Uganda. Is this mainly due to low surveillance activity in the past? Considering other African territories where no RVF has been reported, could this also be due to a lack of surveillance activities?
RVF negative human cases presented in this study had symptoms. What is the likelihood that the observed mortality could be related to other comorbidities? I don't believe this point has been reported. If data is available, it would be interesting to present it.
Tables should be presented differently to provide more clarity.
I believe the tables should be reviewed in more detail, as there is repeated data that is not coincidental. For example, Table 1 shows a total of 61 confirmed cases, of which five are women and 56 are men. However, in Table 2, under the 'Gender' heading, six women and 55 men are listed. Table 1 states that 54 of the positive cases had contact with cattle, whereas Table 2 states that there were 53. In Table 2, 'farmer' and 'butcher' appear in bold by mistake, or is this intended to indicate something?
If animal contact is a risk factor, it is striking that there are no differences in this regard between positive and negative cases. Is there any reasoning to clarify this? Was it possible to monitor the status of animals that came into contact with negative cases? Of the 13 positive cases detected on the farm, how many were negative after having contact with animals that had aborted?
In the phylogenetic tree figures, the font size of the lettering is quite small, so it is difficult to read. I think this could be redone for greater clarity. This might also be desirable for the M and S segment trees (in the supplementary figures).
Have the authors considered depositing the obtained sequences in genomic databases? If so, this is not mentioned in the manuscript.
Reviewer 2 Report
Comments and Suggestions for Authors
This study reports an outbreak investigation of Rift Valley fever virus (RVFV) associated with an abortion storm in dairy cattle, integrating clinical, laboratory, and entomological data. The study has some value from an outbreak response perspective. However, there are several major methodological and interpretive concerns.
Major Comments
- The authors reported RVFV-positive cattle, humans, and mosquito pools, but there is no clear epidemic timeline or geographic/epidemiological linkage presented. A detailed epidemic curve, map of farm and herd distribution, and temporal transmission chainamong animals, humans, and vectors are needed to strengthen outbreak inference.
- While some human samples tested positive for RVFV IgM or RT-PCR, it is unclear whether these individuals had direct contact with infected cattle or whether they were epidemiologically linked to the abortion storm. Without such exposure data, the proposed One Health transmission chainremains speculative.
- There is no distinction made between vaccinated and unvaccinated cattle, making it difficult to interpret whether seropositivity reflects vaccine response or true infection. In addition, important covariates such as gestational age, parity, clinical signs, and barn densitywere not presented, which are critical for understanding risk factors associated with abortion.
- IgM-positive samples were not confirmed using neutralization tests(e.g., PRNT), raising concerns about potential cross-reactivity or false positives. Furthermore, no viral load data (e.g., Ct values or copies/mL) were provided from animal samples to correlate with clinical severity. The analysis is entirely descriptive with no statistical comparison between cases and non-cases (e.g., attack rates or odds ratios).
Minor Comments
1.The manuscript lacks a clear definition of suspected vs. confirmed RVF cases in animals and humans, as well as criteria used to define abortion (e.g., gestational age, placental appearance, pathological features).
2.The conclusion section uses assertive terms such as “confirmed outbreak” and “clear association”. In light of the incomplete evidence, more cautious language such as “suggests” or “likely associated” is recommended.
Round 2
Reviewer 2 Report
Comments and Suggestions for Authors
none